# Primary determinants of water yield services in arid NW China: An empirical analysis of Gansu Province

Hui Yu, Bo Zhang ⓘ *, Qianqian He, Hou Xiao

College of Geography and Environmental Science, Northwest Normal University, Lanzhou, China

* zhangbo@nwnu.edu.cn

## Abstract

Water yield services (WYs) play a crucial role in the hydrological cycle and water resource allocation in terrestrial ecosystems. Therefore, modeling their dynamic variation characteristics and driving mechanisms is of extensive practical significance in guiding ecological management practices in arid and semi-arid regions. Gansu Province is located in the heart of northern China. It is rich in wildlife resources and has numerous ecological reserves, whose ecological transitions profoundly affect the northern region and the ecological security at the national scale. In recent years, Gansu Province has encountered the severe challenge of water resources. It is enduring pollution and a severe imbalance between supply and demand owing to the twofold influence of global warming trends and high-intensity human activities. Based on this, this study quantitatively analyzed the characteristics of the dynamic variation in WYs in Gansu Province using the InVEST model and revealed the key factors driving this dynamic variation. The results show that the WYs in Gansu Province fluctuated between 278.37 and 381.96 mm during 2000–2022, with an average WY of 61.09 mm. The rate of spatial variation in WYs was mainly concentrated between −2 and 5 mm/yr and increased at a rate of 1.41 mm/yr. The spatial heterogeneity of WYs was differentiated significantly by natural and socio-economic influences, with precipitation explaining the highest degree of spatial heterogeneity in WYs ($q = 0.49$–$0.62$) and the strongest interaction between precipitation and actual evapotranspiration ($q = 0.94$). Meanwhile, the interaction between precipitation and land use increased from 0.68 in 2000 to 0.75 in 2022. Moreover, the explanatory power of the interaction between the two showed an increasing trend. In addition, the correlations between each driver and alterations in the WYs showed spatial variations, and the characteristics of each factor differed at different spatial scales. The GDP, proportion of urban construction land, and proportion of arable land area had significant negative spatial effects on WYs. Meanwhile, precipitation had a positive spatial effect on WYs.

**Data availability statement:** All relevant data are within the paper and its Supporting information files.

**Funding:** This research was funded by the National Natural Science Foundation of China (NO. 41561024). The funder (Bo Zhang) not only provided financial support but also played a significant role in conceptualization and handled the writing, review, and editing for this study.

**Competing interests:** The authors have declared that no competing interests exist.

## Introduction

Ecosystem services refer to all the advantages that humans obtain from nature. These broadly encompass the four dimensions of provisioning, regulating, cultural, and supporting services [1,2]. Water yield services (WYs) constitute one of the most critical provisioning services in terrestrial ecosystems [3,4]. WYs are determined by complex ecohydrological processes, including precipitation, runoff, infiltration, and evapotranspiration. At the raster scale, WYs can be conceptually simplified as the residual of precipitation (P) minus actual evapotranspiration (AET) under steady-state conditions (WYs=P−AET), where AET encapsulates both evaporative (E) and transpirative (T) fluxes from vegetated surfaces [5]. However, for comprehensive water budget analysis, WYs can also be expressed as: WYs=precipitation−evapotranspiration−infiltration−water usage+runoff, which explicitly partitions hydrological fluxes into parallel pathways including deep percolation (infiltration), anthropogenic withdrawals (water usage), and lateral flow components (runoff). These formulations are consistent when contextualized by scale: the simplified equation assumes negligible storage changes ($\Delta S \approx 0$), while the expanded version enumerates all major water balance components [6]. The WYs value represents the difference between the average annual precipitation and AET on a raster scale, based on the principle of water balance. They are closely related to regional water allocation and food security, and have become a significant constraint on sustainable development in arid and semi-arid regions. Therefore, in-depth analyses of the temporal and spatial dynamics of WYs and their driving factors hold high practical significance for maintaining ecological security and guiding management practices in these regions.

The formation of WYs is determined by multiple factors, including precipitation, runoff, infiltration, and evapotranspiration. In recent years, large-scale modeling techniques based on remotely sensed data have played a pivotal role in the field of WY assessment and quantification [5,6]. The examples include the Soil and Water Assessment Tool (SWAT), MIKE SHE, and Integrated Valuation of Ecosystem Services and Tradeoffs (InVEST) models. In particular, the InVEST model, which has relatively low raw data requirements and high-quality simulation and evaluation accuracy, has been widely applied and acknowledged worldwide [7,8]. The water yield module of the InVEST model has been deployed successfully in numerous significant regions, notably in the monsoon-influenced hilly watersheds of South China and at the national scale in the United Kingdom [9,10]. These practices validate the effectiveness and practicability of the model and provide strong scientific support for local ecological security and water resource management. Previous studies have determined that climatic factors, particularly precipitation, dominate the spatial and temporal trends of WYs. These factors are constrained by various aspects such as the land use and topography. Pei et al. (2022) determined that climate and land-use variation contributed 88% and 12% [11], respectively, to WYs. Fang et al. (2022) determined that afforestation reduced the supply of WYs, although it increased the carbon stock advantages [12]. Ning et al. (2023) observed that the precipitation variability and its interactions with terrestrial water storage determined the WYs variations [13]. WYs are complex ecohydrological processes under the coupling of multiple factors such

as climate change and human activities. The degree of explanation by different driving factors displays significant regional variability. Most studies have mainly analyzed the influence of natural factors on WYs and were deficient in the spatial differentiation of socioeconomic factors on WYs. Therefore, the evaluation of WYs under multifactor coupling is of high practical significance for a more extensive understanding of the variations in water resource supply in terrestrial ecosystems.

Current studies on the drivers of WYs mainly use correlation analysis, principal component analysis, multiple linear regression analysis, and cluster analysis to evaluate the correlation between WYs and natural and anthropogenic factors [14–16]. However, WYs may be influenced by the interactions of multiple factors during the production process. These effects may be nonlinear or complex [17]. The current research methodology is based on statistical data. The covariance between influencing factors may affect the accuracy of the results. Certain important impact factors may have also been excluded from the analyses because of covariance. Additionally, the spatial heterogeneity and interactions among drivers are difficult to assess, and spatial heterogeneity may have a significant effect on the WYs [18]. However, the conventional linear statistical methods do not address these problems. Geodetectors are a group of statistical methods that detect spatial variability and reveal the underlying driving forces [19]. The geodetector theory is mainly based on the spatial variability of WYs. If a particular influencing factor causes a variation in WYs, these would exhibit a similar spatial distribution and variability as the influencing factor. Therefore, geodetectors can prevent the covariance between data and quantify the effects of single and multiple factors on the spatial distribution of WYs. As a local regression analysis method, the geographically weighted regression (GWR) analysis has been applied to evaluate the factors affecting ecosystem services because it can better reveal the regional differences in influencing factors [20,21]. GWR model introduces an estimation of impacts in different regions. This can reflect the spatial non-stationarity of parameters in different spaces, whereby the relationship between variables can vary with spatial location and the results are more in line with the objective reality [22]. Therefore, in this study, the explanatory capability of driving factors and their spatial role patterns were evaluated extensively from both natural and human activities using the Geodetector and GWR models.

Gansu Province is located in an arid and semi-arid region. It is rich in biodiversity and is an important ecological security barrier in China. However, confronted with the severe challenges of climate change and high-intensity human activities, water supply has become a core ecological issue constraining the ecological security and sustainable development of the region. Water scarcity severely constrains the local sustainable development potential. Additionally, it significantly affects the basic provisioning functions of ecosystems, particularly the WYs of terrestrial ecosystems. Therefore, there is an urgent need to assess the dynamic evolution of local WYs and their driving mechanisms to address water crises. The objectives of this study were to (1) simulate the dynamic evolution characteristics and spatial trends of WYs in Gansu Province during the period 2000–2022 and (2) identify the important driving factors affecting WYs in Gansu Province.

## Materials and methods

### Study area

Gansu Province (92°13′E–108°46′E, 32°11′E–44°57′N) is located in the inland area of Northwest China (Fig 1). Most of the area is located in the second terrace of China. Gansu Province has complex and diverse geomorphological types with plateaus, mountains, and deserts interspersed with multiple geomorphological types. As an important water conservation and recharge area in the upper reaches of the Yangtze and Yellow Rivers, Gansu Province occupies a central position in China's two screens and three belts' ecological security strategy system. The system is a key link between the ecological barriers of the Qinghai–Tibet Plateau and Loess Plateau–Chuan–Yunnan Ecological Barrier. Additionally, it is an important part of the northern China sand belt, and has an irreplaceable and important role in safeguarding the ecological security of the country. It is also an important part of the sand control belt in northern China and plays an irreplaceable and important role in maintaining the national ecological security. Gansu Province is located deep inland in northwest China. It has a typical temperate continental monsoon climate, short summers, and long winters. The average annual temperature ranges from 0 to 14°C, the average annual precipitation ranges from 40 to 760 mm, and the precipitation is mainly concentrated

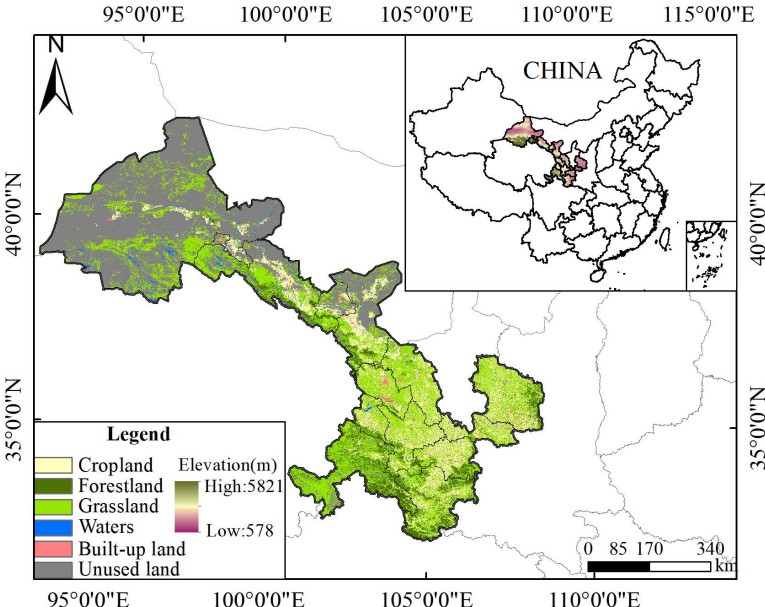

**Fig 1. Schematic diagram of the study area.**

in summer. Arid and semi-arid regions occupy three-quarters of the total area of Gansu Province. The region encounters severe soil erosion and land desertification within the region.

## Data requirement and preparation

As detailed in Table 1, the InVEST model's WY module implementation integrated heterogeneous spatial datasets with varying resolutions (30 m and 1 km) and temporal scales. To ensure geometric consistency across all input layers, we first standardized the coordinate reference system by reprojecting all datasets to the Krasovsky_1940_Albers equal-area projection, which preserves spatial accuracy in our study region's mid-latitude context. Following this preprocessing step, we addressed resolution heterogeneity through a two-stage workflow: (1) retaining the native 30 m datasets for high-resolution feature capture, and (2) resampling the 1 km datasets to match the 30 m grid using nearest-neighbor interpolation. This non-parametric resampling approach maintains original pixel value integrity while subdividing coarse grids into 30 m sub-units, thereby enabling pixel-wise computations without introducing interpolation artifacts. The resulting spatially harmonized datasets ensure both thematic consistency (retention of original measurement values) and geometric fidelity (positional accuracy at 30 m scale), forming a robust analytical foundation for subsequent hydrological modeling processes. Infiltration is not directly calculated as an independent parameter in the InVEST WY module used in this study. However, its effects are implicitly considered through soil property data (e.g., soil texture, water holding capacity) which influence soil infiltration capacity, and through the calculation of AET. These factors collectively affect the water yield calculations by influencing water retention and loss processes in the model.

## InVEST model water yield module

**Quantifying WYs.** WYs are quantified based on the water balance and are essentially the difference between the precipitation and AET at the raster scale [23,24]. The specific calculation process is as follows:

$$Y_{x,j} = \left(1 - \frac{AET(x,j)}{P(x)}\right) \times P(x)$$

(1)

**Table 1. Data sources and pre-processing.**

| Name | Time | Description | Role in Model |
|---|---|---|---|
| Land use | 2000–2022 | Sourced from the Resource and Environmental Sciences Data Centre of the Chinese Academy of Sciences (http://www.resdc.cn). | This is used to characterize the spatial distribution of different land use types, which directly affects the actual evapotranspiration and water yield in the model, as distinct land use categories exhibit varying capacities for evapotranspiration and water production. |
| DEM | 2020 | Sourced from Geospatial Data Cloud (http://www.gscloud.cn/). | This is used to quantify terrain characteristics, which influence precipitation distribution, runoff pathways, and evapotranspiration processes, thereby indirectly affecting water yield. |
| Average annual precipitation | 2000–2022 | National Earth System Science Data Sharing Service Platform (http://www.geodata.cn). | As a primary climatic variable input to the model, it directly influences water yield, forming the fundamental basis for its calculation. |
| Reference crop evapotranspiration | 2000–2022 | Based on PenmanMonteith formula were interpolated into the grid data in ANUSPLINE Software. | As a critical component for calculating actual evapotranspiration, it serves as a pivotal parameter in the model for evaluating water loss, thereby directly influencing water yield outcomes. |
| Soil property data | -- | Harmonized World Soil Database (HWSD), was from the National Cryosphere Desert Center (https://www.crensed.ac.cn/portal/) | This module provides information on soil type, texture, and water holding capacity, which influence soil infiltration capacity and water retention capacity, thereby indirectly affecting water yield and evapotranspiration processes. |
| Sub-basin maps | -- | Based on DEM, these were processed using ArcGIS hydrological analysis tools. | This function is used to delineate sub-basins within the study area, facilitating the analysis of hydrological processes and water yield characteristics across different sub-basins. |
| Zhang-value | -- | The Z value usually ranges between 1 and 30. It was derived from the total water resources in the government water resources bulletin. The simulated depth of water production was verified repeatedly with the measured water resources. The modeled WYs were closest to the measured water resources when Z=5.45. | This parameter is used to adjust the water yield depth in the model, aligning simulated values more closely with field measurements, and serves as an essential calibration parameter for model optimization. |

$$\frac{AET_{x,j}}{P_x} = \frac{1 + \omega_x R_{x,j}}{1 + \omega_x R_{x,j} + 1/R_{x,j}} \tag{2}$$

$$R_{x,j} = \frac{k_{x,j} \times ET_0}{P_x} \tag{3}$$

$$\omega(x) = Z\frac{AWC(x)}{P(x)} + 1.25 \tag{4}$$

where $Y_{x,j}$ is the water yield depth (mm/yr) of class $j$ land-use grid cell $x$; $AET_{x,j}$ is the AET (mm/yr) of class $j$ land-use x; $P_x$ is the precipitation (mm/yr) of grid cell $x$; $R_{xj}$ is the Bydyko dryness index of grid cell $x$; $\omega_x$ is the non-physical parameter of the regional natural climate–soil properties; $Z$ is the Zhang coefficient, with values ranging from to 1–30; and $AWC_x$ is the effective plant water content (mm). A table of biophysical parameters was used to construct the InVEST model (see the S1 Table for details).

**Theil–Sen trend analysis.** The Theil–Sen trend analysis method is a non-parametric slope estimation method. It is less affected by outliers, with a strong noise resistance for long time analysis [25,26]. It is capable of spatially reflecting the trend of the time series variations. The variation slope is calculated as

$$Slope = median\left(\frac{WY_k - WY_i}{k - i}\right) \forall_k > i \tag{5}$$

where *slope* is the slope of the trend of the annual average WYs; $WY_k$, and $WY_i$ are the WYs (mm) in the $k_{th}$ and $i_{th}$ year, respectively; and median is the median function. When *slope* >0, it indicates that the WYs in Gansu Province show an increasing trend on the spatial scale overall. When *slope* < 0, it indicates that the WYs in Gansu Province show a reducing trend on the spatial scale as a whole trend.

**Geodetector.** Geodetectors are commonly used to detect spatial variability and reveal the factor-driven effects. These are widely used in natural, social, and environmental fields [27–29]. In this study, the factor detection and interaction detection in a geodetector were applied to analyze the effects of the influencing factors and their interactions on WYs. Natural geographic features (DEM, slope, precipitation, air temperature, and evapotranspiration) and anthropogenic factors (population density, GDP, and land use) were comprehensively considered as the influencing factors. Finally, eight factors were selected as the independent variables. Factor detection was measured by the q value using the following expression:

$$q = 1 - \frac{\sum\limits_{h=1}^{L} \sum\limits_{i}^{N_h} \left(Y_{hi} - \overline{Y_h}\right)^2}{\sum\limits_{i}^{N} \left(Y_i - \overline{Y}\right)^2} = 1 - \frac{\sum\limits_{h=1}^{L} N_h \sigma_h^2}{N\sigma^2} = 1 - \frac{SSW}{SST}$$

(6)

where $h$ = (1 to $L$) is the number of classifications. $N_h$ and $N$ are the number of sampling units within the stratum of the whole region, respectively. σ and σ are the inter-stratum variance of the whole region, respectively. SST and SSW are the total sum of squares and inner sum of squares, respectively. Q-statistic is a monotonic function of the strength of spatial stratified heterogeneity and $q \in [0,1]$. Its value increased as the intensity of the stratified heterogeneity increased. This indicates that the influencing factor had a more significant effect on WYs.

## Geographically weighted regression model

The OLS model assumes that the degree of influence of the independent variable on the dependent variable is globally homogeneous, i.e., the obtained regression coefficients are spatially constant, ignoring the effect of the spatial location of the data on the degree of action of the independent variable [30]. The GWR model is an extension of the OLS model in that it incorporates information regarding the location of the data in the form of a spatial weighting function into the regression process using neighbors. The GWR model is a typical local regression model. Each sample point is regressed one time. Its formula is as follows:

$$\hat{y}_i = \beta_0 \left(u_i, v_i\right) + \sum_{k=1}^{p} \beta_k \left(u_i, v_i\right) \chi_{ik} + \varepsilon_i$$

(7)

where $y_i$ is the estimated value of the $i_{th}$ point; $(u_i, v_i)$ are the coordinates of the $i_{th}$ point; $\beta_0 (u_i, v_i)$ is the constant term of the $i_{th}$ point; $x_{ik}$ is the $k_{th}$ independent variable of the $i_{th}$ point; $\beta_k (u_i, v_i)$ is the regression coefficient of the $k_{th}$ independent variable of the $i_{th}$ point; $\varepsilon_i$ is the residual term of the model for the $i_{th}$ point; $i$ = 1, 2, 3,..., $n$ ($n$ is the number of sample points); and $k$ = 1, 2, 3,..., $p$ ($p$ is the number of independent variables). In this study, we selected 11 main factors affecting the distribution of WYs: the proportion of forestland land area, proportion of cropland land area, proportion of grassland land area, proportion of built-up land area, population density, GDP, topography, slope, temperature, precipitation, and evapotranspiration. Prior to the implementation of GWR, local covariance diagnosis was performed by OLS to reduce the covariance. Influences with a variance inflation factor (VIF) higher than seven were removed (temperature, slope, and evapotranspiration). The remaining influences (proportion of land area, proportion of cropland area, proportion of grassland area, proportion of built-up land area, population density, GDP, topography, and precipitation) were input into the GWR model.

## Results

### Analysis of climate and land-use dynamic variations

The precipitation, AET, and WYs in Gansu Province fluctuated from 2000 to 2022. During the study period (Fig 2), the minimum precipitation value appeared in 2004 (278.37 mm), the maximum value in 2019 (381.96 mm), and the multiyear average was 315.73 mm. The minimum AET value appeared in 2004 (233.38 mm), the maximum value in 2019 (284.53 mm), and the multiyear average was 254.22 mm. The minimum WYs value appeared in 2015 (38.85 mm), the maximum value in 2021 (114.11 mm), and the multiyear average was 61.09 mm. Spatially, precipitation and AET exhibited a decreasing trend from the southeast to the northwest (Fig 3a, b). This indicated that high precipitation is accompanied by high evapotranspiration in southeastern Gansu Province and that high WYs are mainly concentrated in the southwest of the study area (Fig 3c). The maximum value of WYs (1203 mm) was recorded in the southwest of the study area, and a decreasing trend was observed towards the northeast.

From 2000 to 2020, unused land, grassland, and cropland dominated Gansu Province (Table 2). These accounted for approximately 90% of the total study area. Water and built-up land accounted for approximately 2% of the total study area. During the period 2000–2020, the areas of cropland and unused land decreased by 2.59% and 1.58%, respectively. Furthermore, both showed a significant decrease in precipitation and WYs. During the period 2000–2020, the areas of cropland and unused land decreased by 2.59% and 1.58%, respectively. Furthermore, both showed a continuous downward trend. Built-up land increased by 55.95% and showed a continuous increasing trend. The other three types of land use (forestland, grassland, and water) showed a fluctuating increasing trend. In general, the land-use variations in the study area were highlighted by the significant expansion of construction land. The area variations in the other five land-use types were relatively small.

### Spatial trend analysis of WYs

Trend analyses showed that the precipitation, AET, and WYs in Gansu Province showed spatial trends of a gradual increase from the north to the south. During the study period, the spatial variation rate of precipitation was mainly concentrated in the range of −1–5 mm/yr (Fig 3d), with a linear trend of 2.21 mm/yr across Gansu Province. The increase in precipitation was mainly concentrated in the central and southern parts of Gansu, where the maximum value of 9.80 mm/yr was attained. Meanwhile, the decrease was mainly concentrated in the northwestern part of

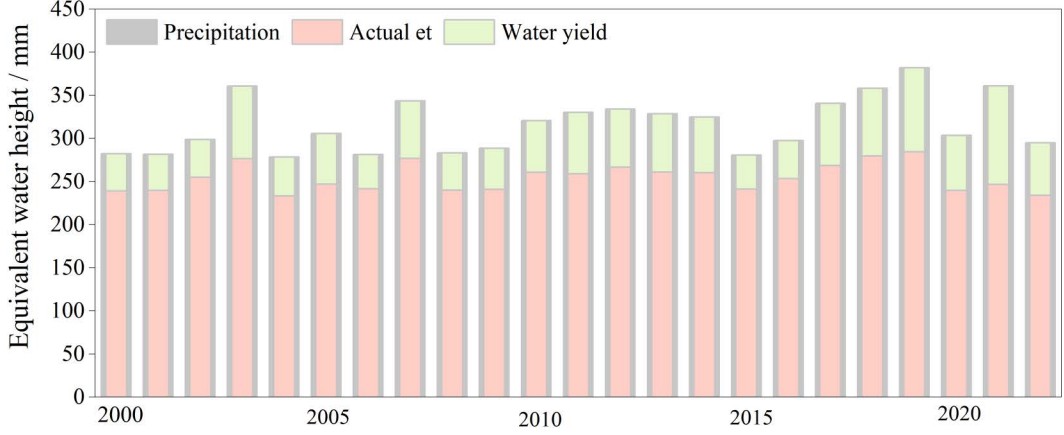

**Fig 2. The multi-year average values of precipitation, actual evapotranspiration, and water yield in Gansu Province from 2000–2022.** Note: Gray represents the annual average precipitation; pink represents actual evapotranspiration; and green represents water yield.

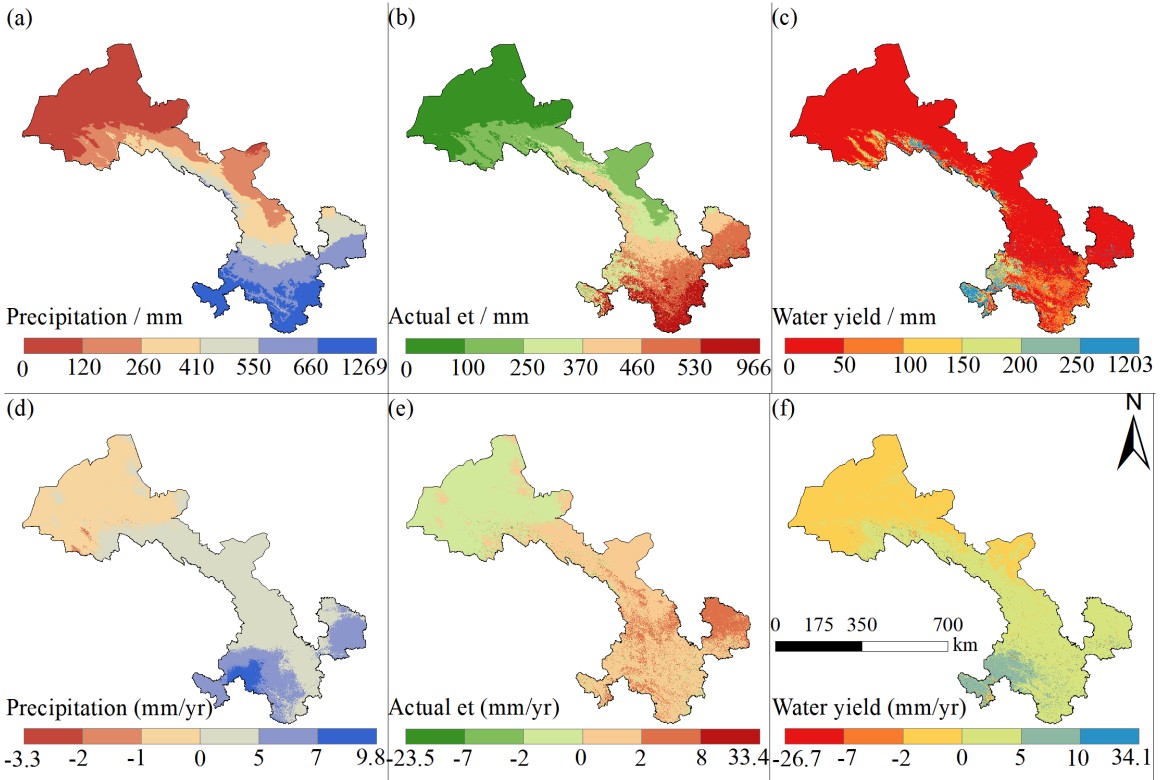

**Fig 3. Spatial distribution of precipitation, AET, and mean WYs (a-c) and analysis of year-by-year trend results (d-e) in Gansu Province, 2000–2022.**

**Table 2. Percentage of area occupied by different land use types in Gansu Province from 2000–2020 (%).**

|  | 2000 | 2005 | 2010 | 2015 | 2020 |
|---|---|---|---|---|---|
| Cropland | 15.42 | 15.39 | 15.29 | 15.25 | 15.02 |
| Forestland | 8.84 | 8.97 | 9.06 | 9.05 | 9.05 |
| Grassland | 33.52 | 33.57 | 33.60 | 33.59 | 33.76 |
| Waters | 0.78 | 0.77 | 0.83 | 0.83 | 0.90 |
| Built-up land | 0.84 | 0.89 | 1.01 | 1.12 | 1.31 |
| Unused land | 40.60 | 40.40 | 40.21 | 40.16 | 39.97 |

the province. The spatial variation rates of AET were mainly concentrated in the range of −2–2 mm/yr (Fig 3e), with a linear trend of 0.53 mm/yr across Gansu Province. The increase in AET was mainly concentrated in the southern region. This indicated that the evapotranspiration in the southern region increased with time and that the dissipation of precipitation and surface runoff increased. The spatial variation rate of WYs was mainly centered between −2–5 mm/yr (Fig 3f), with a linear trend of 1.41 mm/yr across Gansu Province. The increase in WYs was mainly concentrated in the overlapping area between the increase in precipitation and decrease in AET, i.e., in the southern part of the study area. In the central part of the study area, there was a mixed area with both increasing and decreasing WYs. The tradeoff between precipitation and AET had a major influence on the distributional characteristics of this area.

## Analysis of WYs drivers

The magnitudes of the q-values obtained by the geodetector reflect the importance of the influence of natural geographic features or anthropogenic factors on the spatial heterogeneity of the WYs in different periods. The one-way probe results from the geodetector (Fig 4) showed that in 2000, the main influences on the WYs in the study area were of precipitation (0.49), DEM (0.25), slope (0.23), temperature (0.16), and evapotranspiration (0.14). The one-way probe results for 2010 were approximately identical to those for 2000. However, in 2022, precipitation and evapotranspiration began to dominate. Their explanatory powers for the spatial differentiation of the WYs attained 0.62 and 0.27, respectively. Land use had an explanatory power of 0.23 in this period. In general, the explanatory rate for the spatial heterogeneity of WYs by natural factors was high, whereas that for anthropogenic factors was generally low. The preceding content was an analysis of the degree of influence of individual factors on the distribution of WYs. However, in the actual process, complex interactions

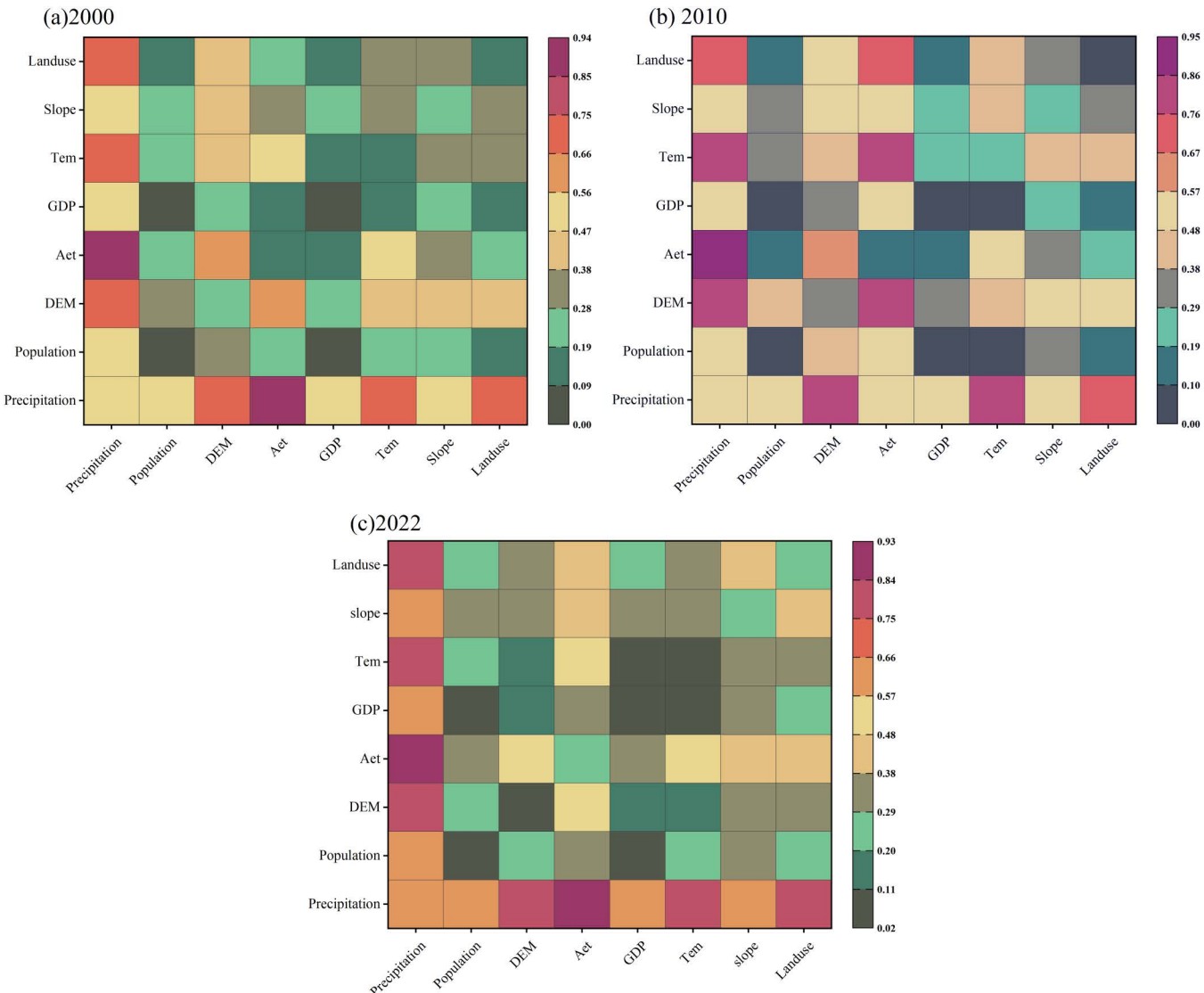

**Fig 4. Interaction between drivers affecting water production and one-way detection results, 2000–2022.**

among multiple factors jointly determine the spatial pattern of WYs. Multifactor interaction probes also demonstrated that the effects of factor interactions on the spatial distribution pattern of WYs were higher than the degree of influence of a single factor, which manifested as a two-factor enhancement and a nonlinear enhancement (Fig 4). In 2000, the interactions between precipitation and AET, DEM, and temperature exceeded 0.7. The highest interaction force for precipitation with AET was 0.94. The results for 2010 and 2022 were generally consistent with those for 2000. Significantly, the interaction between precipitation and land use increased from 0.68 in 2000 to 0.75 in 2022, and the explanatory power of the interaction between the two showed an increasing trend.

The spatial impacts of different drivers on WYs showed heterogeneity, and the influences of the drivers varied across years. The adopted GWR model is highly explanatory of WYs (Fig 5). It can effectively explain the relationship between drivers and WYs. During the period 2000–2022, the cropland area share was negatively correlated with WYs across space. This relationship strengthened gradually from the northwest to the southeast. The proportion of forested land was positively related to WYs in the north and center, and negatively related to WYs in a small part of the south. The proportion of grassland area had a strong positive correlation with WYs in the entire region. This relationship strengthened gradually from the north to the south. In 2000, the proportion of built-up land was negatively correlated with WYs throughout most of the study area, and the growth of built-up land had an adverse impact on WYs. However, by 2010, the two were negatively correlated only in the northern region. In 2022, the proportion of built-up land was negatively correlated with WYs in the north and positively correlated in the south. The elevation was positively correlated with WYs throughout the region, with the strength increasing gradually from the north to the south. The relationship between precipitation and WYs is consistent with those reported in previous studies in that both are positively correlated across the region and that the strength of the relationship increases from the north to the south. Population density was negatively correlated with WYs only in 2000. However, the relationship became positive between 2010 and 2022, with the strength of the relationship increasing gradually from the south to the north. GDP was negatively correlated with WYs throughout the study period, with the strength of the relationship decreasing gradually from the north to the south.

## Discussion

### Spatial distribution pattern of WYs in Gansu Province

The WYs in Gansu Province show spatial differentiation characteristics from the south to the north. These are mainly constrained by a combination of natural and socio-economic factors. The WY value is the difference between the average annual precipitation and AET on a raster scale based on the principle of water balance [31,32]. It is constrained by the constraints of land use. At the spatial scale, the areas of high WYs are those where high precipitation and low AET coincide. This is consistent with the observation that WYs represent a tradeoff between precipitation and AET [33]. Atmospheric precipitation is an important mode of water recharge in terrestrial ecosystems. The capability of precipitation to reach the surface and be converted into runoff is constrained by different land-use types. The southern part of Gansu Province has a relatively low elevation (Fig 1) and is dominated by croplands, woodlands, grasslands, and built-land. The differences in land-use types directly affect the evapotranspiration rate of subsurface and surface cover conditions [34,35]. This, in turn, affects the regional WYs. In 2020, for example, the concentrated distribution area of forest land and construction land was the high-value area of WYs. The two may affect the regional WYs from the following aspects: On the one hand, the forestland in Gansu Province is mainly distributed in the southwestern mountainous area with a relatively high elevation, and the warm and humid airflow is compelled to lift by the topography to straightforwardly form the terrain rain. The dominant wind direction in the southwestern mountainous regions of Gansu, including the Longnan Mountains and Gannan Plateau, changes with season and topography. Summer brings the southeast monsoon, carrying warm, humid air that cools and condenses over the elevated land, producing abundant rain. In winter, the northwest monsoon prevails with drier, cooler air, leading to less rain. Overall, the southeast monsoon dominates in summer, driving topographic precipitation. Simultaneously, affected by the elevation factor, the air temperature of this region is lower than

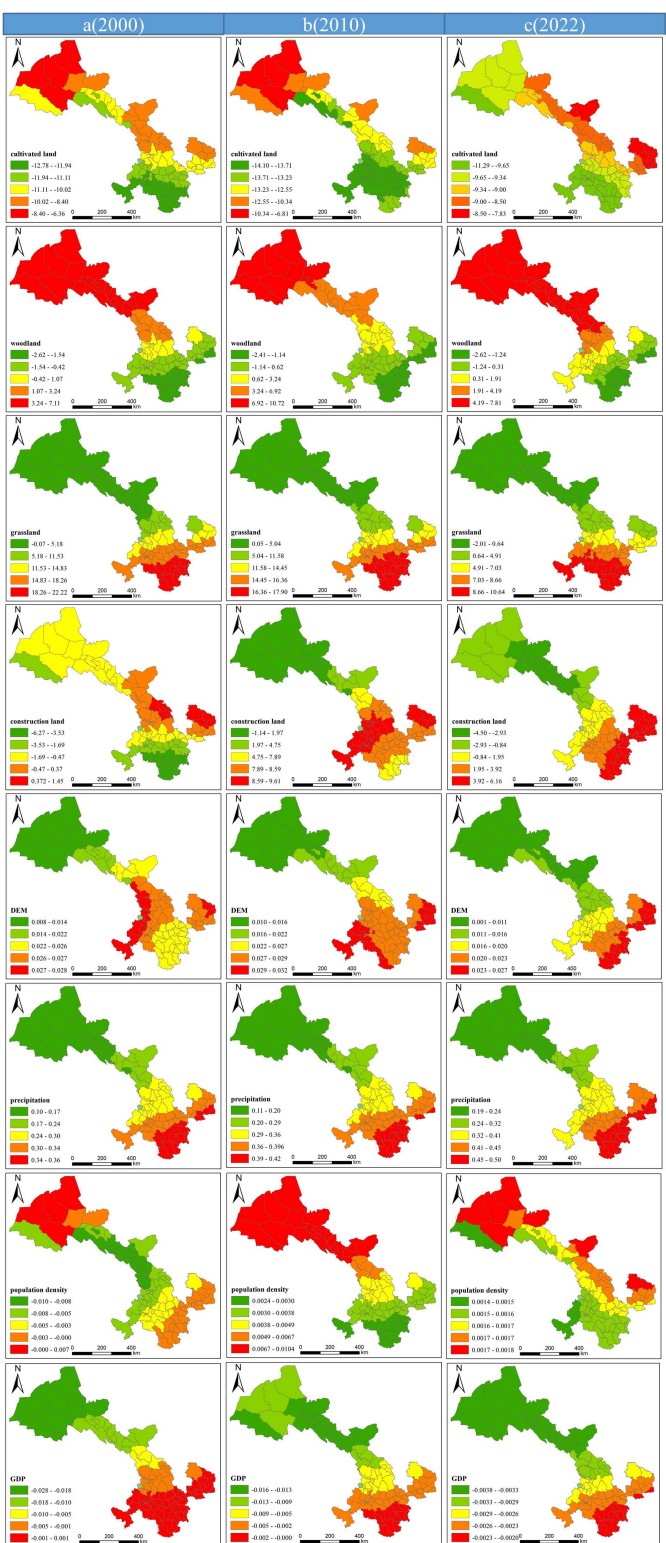

**Fig 5. Spatial distribution of regression coefficients for drivers affecting WYs, 2000–2022.**

that of the plains, and evapotranspiration is relatively less. Consequently, more precipitation is retained in the terrestrial ecosystems. On the other hand, in the built-up land type, the hardening of the ground contributes to the formation of impermeable surfaces on the subsurface [36,37]. The formation of impermeable surfaces reduces the time for infiltration and concentration of output runoff from terrestrial ecosystems, and more surface runoff and flood peaks are gathered in a short period of time. The low-value areas of WYs are mainly concentrated in the northwestern part of the study area, which is dominated by unused land (Table 3). The northwestern part of Gansu Province is a low-precipitation area with low vegetation cover. The main form of evaporation is soil evapotranspiration, which is characterized by high infiltration. Thus, the distribution area of unused land in Gansu Province is called the low-value area of WYs. Gansu Province is located in the northern hinterland of China. The dynamic evolution of its water resource supply is of strategic importance for ecological protection and water resource management. In this study, the spatial distribution pattern of WYs in Gansu Province was clarified. This is of high practical significance for the rational spatial allocation of water resources and water resource protection. Throughout the study period (2000–2022), the WYs in Gansu Province showed an increasing spatial trend and were mainly concentrated in the southwestern part of the study area, where forestland is widespread. To a certain extent, this indicates that the implementation of ecological protection policies in Gansu Province has achieved substantial results and that the stability of the ecosystem has increased. Therefore, in the future, the implementation of ecological protection policies and awareness of water resource protection should be continued to gradually improve the status quo of water resource scarcity in the study area.

## Attribution analysis of the dynamic evolution characteristics of WYs

The results of both geodetector and geographically weighted regression models indicate that 1) natural factors are the main factors affecting the spatial distribution of WYs and 2) the explanatory power of natural factors is generally higher than that of anthropogenic factors. Natural factors such as precipitation, evapotranspiration, and DEM affect the spatial distribution of WYs in Gansu Province. This study determined that the spatial distribution characteristics of precipitation, AET, and WYs were essentially identical. With regard to the spatial distribution of WYs, the high-value area is concentrated in the high-altitude area in the southern part of Gansu Province, which has higher precipitation and evapotranspiration. The geodetector also determined that the interaction between the two had the highest explanatory power for the spatial distribution of WYs. In general, the higher the temperature, the more vigorous is the evapotranspiration and faster is the water circulation. The higher the humidity, the higher are the precipitation and the relative increase in WYs [38].

According to the regression coefficients of the geographically weighted regression model, precipitation had a significant positive effect on the WYs over almost the entire region. The geodetector results also showed that the interaction between precipitation and land-use types explained the spatial differences in the WYs effectively. This is consistent with previous studies [39–41]. WYs are affected by a combination of climatic and land-use/overburden variations [42]. Climate change can affect the WYs by altering precipitation and evapotranspiration (solar radiation, temperature, and precipitation) [43]. Land-use variations can alter the watershed water cycle, thereby affecting evapotranspiration and infiltration processes and, thus, WYs [44,45]. Grassland and cropland are the main land types in the study area. These exert significant effects on the spatial distribution of the WYs. The effect of grasslands on WYs is mainly reflected in their capability to increase WYs and improve the water-use efficiency [46]. Grassland vegetation releases large amounts of water vapor into the atmosphere through transpiration, thereby forming an important source of precipitation [47]. Meanwhile, the plant root systems on grasslands can infiltrate and store rainwater, thereby effectively reducing runoff [48]. The geographically weighted

**Table 3. Depth of water yield in 2020 under different land-use types (mm).**

| WY/Year | Cropland | Forestland | Grassland | Waters | Built-up land | Unused land |
|---|---|---|---|---|---|---|
| 2020 | 74.51 | 142.19 | 95..01 | 10.56 | 178.46 | 11.90 |

regression model also verified a positive correlation between grasslands and WYs. Although the Gansu Province has a high proportion of arable land, it experiences water scarcity. A study estimated that cropland expansion increased the AET by 182 mm/yr [49]. Because of the insufficient precipitation in the study area, evapotranspiration is higher in the cropland area, and WYs reduce accordingly.

The ground surfaces in urban ecosystems are generally composed of substances such as concrete, asphalt, and cement. These form impervious surfaces, which are vulnerable to the rapid formation of runoff after precipitation reaches the impervious surface [11]. This, in turn, reduces water infiltration and, thus, increase the WYs. In arid areas, even with low population densities, WYs may be limited because of the harsh natural conditions. In contrast, in humid areas, abundant precipitation ensures higher WYs even when the population density is high. Socioeconomic factors (population density and GDP) also strongly influence the spatial differentiation of the WYs in Gansu Province. The GDP growth implies an increase in economic activity, which usually causes an increase in the demand for water resources. For example, large amounts of water are required for industrial production, agricultural irrigation, and urban water supply. As the economy grows, these demands increase further, thereby causing a relative decrease in WYs. Overall, climate change has a higher effect on WYs, whereas land-use/cover variation has a lesser effect. Climatic elements are mainly controlled by natural conditions. Meanwhile, anthropogenic impacts on precipitation indirectly affect WYs, mainly through variations in land use.

## Shortcomings and outlook

First, the WYs module of the InVEST model is based on the principle of water balance. Moreover, it constructs the model framework on a raster scale, which focuses on revealing the differences in the spatial distribution of precipitation and AET. In general, WYs are larger in areas with high precipitation and low AET, and the module focuses on the role of climatic factors. However, WYs are also affected by various factors such as agricultural irrigation and the soil moisture content. The InVEST model simplifies the complexity of the water supply in terrestrial ecosystems by simulating WYs. This causes the simulation results to deviate from the actual water supply situation. Second, given the deficiency of high-quality continuous time-series land-use data, this study used land-use data at a fixed point in time to represent the variations over a five-year period. However, land-use variation is essentially a continuous evolutionary process, and this consideration may have underestimated the actual impact of land-use variation on WYs. Thus, this impact is not fully reflected in the variations in WYs. To further improve the accuracy of WY modeling, future research should aim to improve the temporal resolution of land-use data. Finally, the dynamic evolution of the WYs is the result of a nonlinear combination of natural and socioeconomic factors. In this study, we focused on resolving the effects of several key natural factor indicators on the WYs. Future studies should appropriately consider the socioeconomic factors and expand the scale of the present study. To conclude, a more comprehensive evaluation of WYs and analysis of the drivers is a direction that future research needs to strive to improve.

## Conclusions

In this study, the spatial and temporal evolution characteristics of WYs in Gansu Province were simulated using the WYs module of the InVEST model based on long time-series data from 2000 to 2022. The effects of natural and socioeconomic factors on the spatial differentiation of WYs were resolved using the geodetector and GWR models. The main conclusions are as follows. The land-use types in Gansu Province are dominated by unused land, grassland, and cropland. These account for approximately 90% of the total study area. During the study period, built-up land experienced significant expansion. The high values of WYs in Gansu Province were mainly concentrated in the southwestern part of the study area. These showed an overall decreasing trend from the southwestern to the northeastern part of the study area and an overall increasing trend at a spatial rate of 1.41 mm/yr. The analysis of driving variations showed that the key drivers affecting the variations in WYs in Gansu Province include the precipitation; AET; DEM; and proportions of built-up land, cropland, and forested land. Climate change is the main driver of the variations in WYs. The effect of land-use variation

on WYs was dominated by grassland. It was followed by cropland, forestland, and urban built-up land. The effects of each driver differed spatially. The results of this study scientifically support the formulation of regional water resource policies, socioeconomic development plans, and other future development plans for Gansu Province.

## Supporting information

**S1 Table. The biophysical parameters table utilized in the InVEST water yield model.**
(DOCX)

**S2 File. Manuscript data.**
(ZIP)

## Author contributions

**Conceptualization:** Hui Yu, Bo Zhang.

**Data curation:** Hui Yu.

**Funding acquisition:** Bo Zhang.

**Methodology:** Qianqian He.

**Software:** Hui Yu, Bo Zhang, Qianqian He, Hou Xiao.

**Writing – original draft:** Hui Yu.

**Writing – review & editing:** Bo Zhang.

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
