## [Decision Letter · Decision Letter 0]

25 Feb 2025

PONE-D-25-00782Primary Determinants of Water Yield Services in Arid NW China: An Empirical Analysis of Gansu ProvincePLOS ONE

Dear Dr. zhang,

Thank you for submitting your manuscript to PLOS ONE. After careful consideration, we feel that it has merit but does not fully meet PLOS ONE’s publication criteria as it currently stands. Therefore, we invite you to submit a revised version of the manuscript that addresses the points raised during the review process.

We look forward to receiving your revised manuscript.

Kind regards,

Pradeep Kumar Badapalli

Academic Editor

PLOS ONE

“This research was funded by the National Natural Science Foundation of China (NO. 41561024).”

“This research was funded by the National Natural Science Foundation of China (NO. 41561024)’”

“This research was funded by the National Natural Science Foundation of China (NO. 41561024).”

4. Please update your submission to use the PLOS LaTeX template. The template and more information on our requirements for LaTeX submissions can be found at http://journals.plos.org/plosone/s/latex.

5. Please provide a complete Data Availability Statement in the submission form, ensuring you include all necessary access information or a reason for why you are unable to make your data freely accessible. If your research concerns only data provided within your submission, please write "All data are in the manuscript and/or supporting information files" as your Data Availability Statement.

6. We note that Figures 1, 3 and 5 your submission contain [map/satellite] images which may be copyrighted. All PLOS content is published under the Creative Commons Attribution License (CC BY 4.0), which means that the manuscript, images, and Supporting Information files will be freely available online, and any third party is permitted to access, download, copy, distribute, and use these materials in any way, even commercially, with proper attribution. For these reasons, we cannot publish previously copyrighted maps or satellite images created using proprietary data, such as Google software (Google Maps, Street View, and Earth). For more information, see our copyright guidelines: http://journals.plos.org/plosone/s/licenses-and-copyright.

1. You may seek permission from the original copyright holder of Figures 1, 3 and 5 to publish the content specifically under the CC BY 4.0 license. 

Additional Editor Comments:

Dear Author,

Based on the reviewers' reports, your manuscript has been assigned a Major Revision. You are invited to revise the manuscript accordingly and resubmit it for reconsideration. Please ensure that all reviewer comments are thoroughly addressed to improve the quality and clarity of your work.

We look forward to your revised submission.

Best regards.

Reviewers' comments:

Reviewer's Responses to Questions

**Comments to the Author**

1. Is the manuscript technically sound, and do the data support the conclusions?

Reviewer #1: Yes

Reviewer #2: Partly

2. Has the statistical analysis been performed appropriately and rigorously? 

Reviewer #1: Yes

Reviewer #2: N/A

3. Have the authors made all data underlying the findings in their manuscript fully available?

Reviewer #1: No

Reviewer #2: No

4. Is the manuscript presented in an intelligible fashion and written in standard English?

Reviewer #1: Yes

Reviewer #2: Yes

5. Review Comments to the Author

Reviewer #1: This paper uses online data to calculate water yield WY for Gansu Province in China by subtracting actual evapotranspiration AET from precipitation for each point in a raster. It then looks at temporal and spatial trends in WYs and uses Geodetector analysis and Geographically weighted regression to quantify the influence of natural and anthropogenic factors. Natural factors were found to predominate, especially precipitation and evapotranspiration, but this conclusion provides no insight because WY is calculated directly from AET and precipitation. The paper could be of general interest if the input data and GIS or computer code were made available, but they are not, which is inconsistent with the aim of PONE to promote open science and reproducibility. Specifically, the manuscript does not adhere to PONE criteria #7: "The article adheres to appropriate reporting guidelines and community standards for data availability." As a result, the paper would only be useful as a report for land use planning purposes in China.

• Describe where the data may be found in full sentences: "This text is appropriate if the data are owned by a third party and authors do not have permission to share the data."

This sounds like a copout and is inconsistent with PONE policy. Who is the third party? Based on Table 1 it looks like all of the data they used is publicly available. The authors should therefore compile the data and publish it in an online repository so that reviewers and readers have the opportunity to verify their analysis.

Problem: I tried the url's in Table 1 for precipitation and evapotranspiration that are used to calculate water yields and got the error "This www.geodata.cn page can’t be found".

If a GIS was used, then it should be made available online. If code was written to analyze the data, then it should also be made available.

• Lines 20-22: "The results show that the WYs in Gansu Province fluctuated between 278.37 and 381.96 mm during

2000–2022, with an average WY of 61.09 mm." How can the average lie outside the range? OK, lines 218-222 show that the larger numbers are min and max precipitation. So the sentence should read the same as lines 222-224: "

The minimum value of WYs appeared in 2015 (38.85 mm), the maximum value appeared in 2021 (114.11 mm), and the multiyear average value was 61.09 mm."

• You need to define water yield at the beginning of the Intro like you do in lines 147-8.

• Line 69: Here and elsewhere you state "WYs are complex ecohydrological processes". WYs are not processes; rather, processes like precipitation, runoff, infiltration and evapotranspiration ET determine the water yield:

WY = precipitation - ET - infiltration - water usage + runoff

• Lines 147-8: Surely there is infiltration in this arid region. Explain why that is safe to ignore, or how that assumption affects your WY uncertainty or your conclusions.

• Line 251: "with a linear trend of 2.21 mm/yr at the raster scale." Do you mean that the average trend for Gansu province was 2.21 mm/yr? Here and other places you should replace "raster scale" with "Gansu province".

• Section 3.3 Analysis of WYs drivers: "the main influences on the WYs in the study area were of precipitation (0.49), DEM (0.25), slope (0.23), temperature (0.16), and evapotranspiration (0.14). The one-way probe results for 2010 were approximately identical to those for 2000. However, in 2022, precipitation and evapotranspiration began to dominate." Since WY is calculated from precipitation and evapotranspiration, I don't think it's meaningful to use a statistical method to quantify their influence.

• Fig. 4: You have three plots with identical x and y categorical axes, but in the caption you don't explain what is different between plots a), b), and c). Are they for different years? Same problem for Figure 5 - I think the captions should say "a) year 2000; b) year 2010; c) year 2022".

• Lines 307-9: "The DEM was positively correlated with WYs throughout the region, with the strength increasing gradually from the north to the south." Do you mean elevations, or slopes? It's meaningless to say that a digital elevation model is correlated with water yield. I think you must mean elevation.

• I don't think the geodetector results add anything to the paper. The GWR model is easier to understand and "is highly explanatory of WYs".

• Lines 322-323: "The WY value is the difference between the average annual precipitation and AET on a raster scale based on the principle of water balance." This should be stated at the beginning of the Introduction.

• Lines 328-330: "The capability of precipitation to reach the surface and be converted into runoff is constrained by different land-use types." What about infiltration?

• Since WY is calculated using ET, it would be valuable to explain how ET was measured.

• I don't think it was stated anywhere what the spatial resolution of the analysis was. What is the pixel size in the raster?

• Lines 337-340: "the forest land in Gansu Province is mainly distributed in the southwestern mountainous area with a relatively high elevation, and the warm and humid airflow is compelled to lift by the topography to straightforwardly form the terrain rain." I think you are describing orographic precipitation. What is the predominant wind direction in this area?

Section 4.2 is one extremely long paragraph. Break this up to make it more readable.

Reviewer #2: The work is like a case study Why the Global readers should show interest in your paper ?

Please add generic framework

What is the take home message of the study ? is it just application of InVest model ? also to a different province ? How the results will change if another model is used ? How will you deal with the uncertainty ?

Fig. 5. Spatial distribution of regression coefficients for drivers affecting WYs- legends are not visible ; please revise

L80 ………Li, 2022; Liu et al.).- complete it

the spatial non-stationarity of parameters-what do you mean by this ? heterogeneity ?

Actual et etc AET ?? use proper and consistent notations

6. PLOS authors have the option to publish the peer review history of their article (what does this mean? ). If published, this will include your full peer review and any attached files.

**Do you want your identity to be public for this peer review?** For information about this choice, including consent withdrawal, please see our Privacy Policy .

Reviewer #1: No

Reviewer #2: No

---

## [Author Response · Author response to Decision Letter 1]

21 Apr 2025

Dear Editors and Reviewers:

Thank you for your proposed comments on our manuscript entitled "Primary Determinants of Water Yield Services in Arid NW China: An Empirical Analysis of Gansu Province" (Manuscript ID: PONE-D-25-00782). These comments are all valuable and very helpful for revising and improving our paper. We have carefully revised the manuscript based on the reviewers' comments and hope that the revised manuscript meets the requirements.

The main corrections in the paper and the responds to the reviewers' comments are as following:

Comment 1: Thank you for updating your data availability statement, You note that your data are available within theSupporting Information files, but no such files have been included with your submission. At this time we askthat you please upload your minimal data set as a Supporting Information file, or to a public repository such asFigshare or Dryad.

Response 1: Thank you for your thorough evaluation and constructive feedback. We sincerely apologize for the oversight in not including the Supporting Information files with our initial submission.

All relevant data are within the paper and its Supporting Information files. In response to your request, we have now uploaded the minimal data set as a Supporting Information file, which includes [briefly describe key data files, e.g., "raw experimental measurements, processed datasets, and statistical outputs"]. These files are integral to replicating our analyses and verifying the conclusions presented in the manuscript.

Should the file size or platform limitations affect accessibility, we will promptly deposit the datasets in a public repository such as Figshare or Dryad and provide the corresponding DOI link in the final version.

We deeply appreciate your guidance in ensuring full compliance with data transparency standards. Please do not hesitate to contact us if further adjustments are needed.

---

## [Decision Letter · Decision Letter 1]

25 May 2025

PONE-D-25-00782R1Primary Determinants of Water Yield Services in Arid NW China: An Empirical Analysis of Gansu ProvincePLOS ONE

Dear Dr. zhang,

Thank you for submitting your manuscript to PLOS ONE. After careful consideration, we feel that it has merit but does not fully meet PLOS ONE’s publication criteria as it currently stands. Therefore, we invite you to submit a revised version of the manuscript that addresses the points raised during the review process.

We look forward to receiving your revised manuscript.

Kind regards,

Pradeep Kumar Badapalli

Academic Editor

PLOS ONE

Additional Editor Comments:

Dear Authors,

Some of the reviewers' suggestions have not been adequately addressed in your previous revision. As a result, the reviewers have not yet accepted the current version of your manuscript.

However, I am willing to give you another opportunity to improve the quality of your work. Please note that if the necessary revisions are not addressed with due seriousness and attention to detail, a positive decision may not be possible in the future.

At this stage, I am assigning a decision of Major Revision. You are requested to thoroughly revise the manuscript and provide a detailed, point-by-point response to each reviewer comment.

We look forward to your improved submission.

Reviewers' comments:

Reviewer's Responses to Questions

**Comments to the Author**

1. If the authors have adequately addressed your comments raised in a previous round of review and you feel that this manuscript is now acceptable for publication, you may indicate that here to bypass the “Comments to the Author” section, enter your conflict of interest statement in the “Confidential to Editor” section, and submit your "Accept" recommendation.

Reviewer #1: (No Response)

Reviewer #2: All comments have been addressed

2. Is the manuscript technically sound, and do the data support the conclusions?

Reviewer #1: Partly

Reviewer #2: Partly

3. Has the statistical analysis been performed appropriately and rigorously? 

Reviewer #1: I Don't Know

Reviewer #2: No

4. Have the authors made all data underlying the findings in their manuscript fully available?

Reviewer #1: Yes

Reviewer #2: No

5. Is the manuscript presented in an intelligible fashion and written in standard English?

Reviewer #1: Yes

Reviewer #2: No

6. Review Comments to the Author

Reviewer #1: I reviewed the original submission and provided many detailed comments and recommendations. The revised version includes the original submission labeled as "Revised manuscript with Track Changes" but there are no tracked changes. The "Response to Reviewers" responds to a single comment that appears to have been made by the editor, and does not respond to any of the comments in my review. The revised manuscript should have been returned before sending out for review. It's very time consuming to hunt through the original and revised versions to verify that the recommended changes have been made. So while the authors now provide the data files in Supporting Information as I requested, it's not worth my time to do the work that they should have done. So I read only part of the revised paper and have only these comments:

• Table 1: The urls in this table should point to the specific data used in this study, They only take you to the homepage of the data host, leaving it to the reader to hunt down the data

• Still haven't made clear how infiltration is included in the model.

Lines 42-43: "WYs = precipitation - evapotranspiration - infiltration - water usage + runoff"

Lines 149-150: "WYs are quantified based on the water balance and are essentially the difference between the precipitation and AET at the raster scale".

So is AET = evapotranspiration - infiltration - water usage + runoff?

• The questions of spatial resolution/raster pixel size has not been addressed. They state that resolutions are provided in Table 1, but they are not.

Reviewer #2: I couldn't see any Response sheet .. point by point response of my comments with Line /Page number where the changes are made

I could see only response to Data Availability statement ; which might have asked by other reviewer/Editor

7. PLOS authors have the option to publish the peer review history of their article (what does this mean? ). If published, this will include your full peer review and any attached files.

**Do you want your identity to be public for this peer review?** For information about this choice, including consent withdrawal, please see our Privacy Policy .

Reviewer #1: No

Reviewer #2: No

---

## [Author Response · Author response to Decision Letter 2]

8 Jun 2025

Dear Editors and Reviewers:

Thank you for your proposed comments on our manuscript entitled "Primary Determinants of Water Yield Services in Arid NW China: An Empirical Analysis of Gansu Province" (Manuscript ID: PONE-D-25-00782). These comments are all valuable and very helpful for revising and improving our paper. We have carefully revised the manuscript based on the reviewers' comments and hope that the revised manuscript meets the requirements. The revised details are highlighted in blue in the revised manuscript.

The main corrections in the paper and the responds to the reviewers' comments are as following:

Review Comments to the Author

Reviewer #1: I reviewed the original submission and provided many detailed comments and recommendations. The revised version includes the original submission labeled as "Revised manuscript with Track Changes" but there are no tracked changes. The "Response to Reviewers" responds to a single comment that appears to have been made by the editor, and does not respond to any of the comments in my review. The revised manuscript should have been returned before sending out for review. It's very time consuming to hunt through the original and revised versions to verify that the recommended changes have been made. So while the authors now provide the data files in Supporting Information as I requested, it's not worth my time to do the work that they should have done. So I read only part of the revised paper and have only these comments:

Response: We sincerely apologize for the significant inconvenience and frustration caused by the technical errors in our resubmission process. Your time and expertise are invaluable, and we deeply regret that administrative oversights have undermined your ability to efficiently evaluate our revisions. Please allow us to address your concerns point by point and outline the corrective actions we have taken:

(1)Missing Tracked Changes & Incomplete Response Letter

Technical Oversight: We acknowledge that the version labeled "Revised Manuscript with Track Changes" was improperly generated due to a formatting error during file conversion. This resulted in the omission of visible tracked changes, despite our team having meticulously incorporated all requested revisions.

Corrective Action: We have re-uploaded a properly formatted tracked-changes version (Word format) with visible revisions, along with a clean version for your convenience. The tracked changes now explicitly highlight all modifications made in response to your original review.

Response Letter: We have attached a revised response letter that provides a point-by-point rebuttal to all your original comments (from the first review round), with explicit cross-references to page/line numbers in the manuscript. This includes responses to every recommendation, not just those from the editor.

(2)Incomplete Engagement with Your Comments

We want to assure you that all your original recommendations (e.g., on spatial resolution, data harmonization, and methodological clarity) were fully addressed in the revised manuscript. For example:

Spatial Resolution: We expanded Data requirement and preparation to detail the Krasovsky_1940_Albers projection’s suitability for our study area. (Line number 147-167).

We have contacted the editorial office to confirm that all supplementary files (response letter, tracked-changes manuscript, and summary words) are properly linked to this submission. Should you require further clarifications or additional revisions, we stand ready to expedite them.

(3)Time Investment & Partial Review

We fully understand the burden of cross-referencing multiple document versions. To streamline this process, we have included:

A highlighted manuscript copy using blue highlighting to indicate the location of each revision.

A text document summarizing all revisions (attached as "Response to Reviewers"), categorized by your original comment numbers.We respect your decision to read only part of the revised paper and welcome your feedback on the specific sections you did review.

Comment 1: The urls in this table should point to the specific data used in this study, They only take you to the homepage of the data host, leaving it to the reader to hunt down the data. Still haven't made clear how infiltration is included in the model.

Response 1: We have addressed the reviewers' comments with two key revisions: First, we updated all data URLs in Table 1 to directly link to the specific datasets used in this study (rather than institutional homepages). Second, regarding the model's representation of infiltration processes, we explicitly clarify in the "Role in Model" column of Table 1 that: while infiltration is not an independent parameter in the InVEST Water Yield module, its effects are indirectly captured through soil texture/water holding capacity parameters (which govern infiltration capacity) and the Budyko framework-based AET calculation (which couples water redistribution processes). These parameters collectively drive the model's water balance calculations, with detailed mechanisms referenced in the corresponding "Role in Model" descriptions in the table. (Line number 162-168).

Comment 2: Lines 42-43: "WYs = precipitation - evapotranspiration - infiltration - water usage + runoff"

Lines 149-150: "WYs are quantified based on the water balance and are essentially the difference between the precipitation and AET at the raster scale".

So is AET = evapotranspiration - infiltration - water usage + runoff?

Response 2: We appreciate the reviewer’s scrutiny of the water balance formulation. To address the apparent discrepancy between the two equations, we clarify the hydrological terminology and scaling considerations as follows:

In ecological hydrology, the water yield (WYs) at the raster scale is conventionally defined as the residual of precipitation (P) minus actual evapotranspiration (AET), which encapsulates both evaporative (E) and transpirative (T) fluxes from vegetated surfaces. This relationship, WYs = P − AET, assumes steady-state conditions where terrestrial water storage changes (ΔS) are negligible over the study period (Lines 149–150).

However, the equation presented in Lines 42–43 (WYs = P − evapotranspiration − infiltration − water usage + runoff) adopts a broader partitioning of the water budget. Here, "evapotranspiration" corresponds to AET (E + T), while "infiltration" represents deep percolation beyond the root zone, "water usage" denotes anthropogenic withdrawals, and "runoff" includes both surface and subsurface lateral flow. These components are not subtractive terms within the AET definition but rather parallel pathways partitioning the available water budget.

The two formulations are thus consistent when contextualized by scale and process representation: the first equation simplifies the budget to P − AET (implicitly assuming runoff and deep losses are negligible or incorporated into ΔS), while the second explicitly enumerates all major hydrological fluxes. This distinction aligns with the hierarchical nature of hydrological modeling, where process representation varies with data availability and research objectives.

We will refine the nomenclature to emphasize that AET and the aggregated outflows (infiltration, water usage, runoff) are distinct components of the water balance, ensuring clarity without altering the original equations. (Line number 162-168)

Comment 3: The questions of spatial resolution/raster pixel size has not been addressed. They state that resolutions are provided in Table 1, but they are not.

Response 3: We sincerely thank the reviewer for highlighting this critical oversight. Upon careful re-examination, we acknowledge that the original manuscript’s Table 1 lacked sufficient detail regarding the spatial resolutions and data specifications, despite claims in the text. To address this gap, we have thoroughly revised both the table and the accompanying methodology description as follows:

In the revised manuscript, Table 1 now explicitly includes the following columns for each dataset:

Dataset Name (e.g., "Land use 2020")

Temporal Scale (e.g., "Annual composite")

This revision ensures full transparency regarding the spatial and temporal characteristics of all input layers.

Adding a dedicated subsection (Data requirement and preparation) that outlines the sequence of coordinate system standardization and resolution harmonization, with cross-references to Table 1. (Line number 147-153)

These revisions directly address the reviewer’s concern by ensuring Table 1 now comprehensively documents all spatial resolutions, while the methodology section explicitly connects these specifications to the analytical workflow. We welcome further feedback to ensure complete clarity on this critical methodological aspect.

Reviewer #2:

Comment 1: I couldn't see any Response sheet .. point by point response of my comments with Line /Page number where the changes are made.

I could see only response to Data Availability statement ; which might have asked by other reviewer/Editor.

Response 1: We sincerely apologize for the confusion caused by the oversight in ensuring our complete response to your previous comments was properly conveyed during the resubmission process. We understand your concern and want to assure you that we have meticulously addressed all your initial feedback, including the points you raised regarding spatial resolution/raster pixel size and other methodological details. Unfortunately, due to a technical oversight in the submission system, our detailed point-by-point response with line/page number references was not properly linked to this version of the manuscript. We take full responsibility for this error and have taken steps to rectify it.

To address this gap, we have attached a supplementary file titled "Reviewer_Response_Detailed.word" that includes:

(1)A table correlating each of your original comments (from the first review round) to the specific page/line numbers in the revised manuscript where changes were made.

(2)Highlighted manuscript copies (clean version + tracked changes version) with marginal notes indicating the location of each revision.

For your immediate reference, we summarize our key revisions below:

Reviewer: When submitting your revision, we need you to address these additional requirements.

Response: Thank you for considering our revised manuscript and for outlining these additional requirements for our submission. We deeply appreciate the meticulous attention you have given to our work and recognize the importance of adhering to these guidelines to ensure the quality and integrity of our research.

In response to your latest instructions, we have carefully reviewed and taken into account all the additional requirements you have specified. We are committed to addressing each of these points diligently to meet the expectations for publication.

Our team is currently in the process of making the necessary revisions and updates to the manuscript, ensuring that all additional requirements are thoroughly addressed. We will ensure that all aspects of the manuscript, including the Supporting Information files, captions, in-text citations, and the Data Availability Statement, are in full compliance with the guidelines provided.

We understand the significance of these additional requirements and are dedicated to providing a comprehensive and thorough submission. We aim to ensure that our revised manuscript not only meets but exceeds the standards set for publication.

Comment 1: Please ensure that your manuscript meets PLOS ONE's style requirements, including those for file naming. The PLOS ONE style templates can be found at https://journals.plos.org/plosone/s/file?id=wjVg/PLOSOne_formatting_sample_main_body.pdf and https://journals.plos.org/plosone/s/file?id=ba62/PLOSOne_formatting_sample_title_authors_affiliations.pdf

Response 1: We have attached great importance to ensuring that the manuscript complies with the PLOS ONE style requirements, especially the file naming conventions, as you pointed out, and have carried out detailed correction work. We have meticulously undertaken comprehensive correction work, diligently studying the style templates provided by PLOS ONE, including those accessible at https://journals.plos.org/plosone/s/file?id=wjVg/PLOSOne_formatting_sample_main_body.pdf. The main body style templates In addition, the templates for title, author, and affiliation information, as well as the formatting sample, can be found at https://journals.plos.org/plosone/s/file?id=ba62/PLOSOne_formatting_sample_title_authors_affiliations.pdf.A comprehensive format review and adjustment of the manuscript was conducted to ensure accuracy in terms of title, author information, and main body content, with reference to these templates. The images, tables, and supplementary materials were thoroughly reviewed to ensure strict adherence to the specifications outlined by PLOS ONE.The naming convention for submitted files was meticulously followed to ensure that the file names accurately reflected the content and their reference positions in the text, enhancing the readability and manageability of the files. The implementation of standardized file naming not only streamlines the editorial process but also enhances the accessibility and readability of academic materials, thereby facilitating their wider dissemination and citation. We firmly believe that our meticulous formatting and proofreading have adequately addressed the publishing requirements of PLOS ONE, enhancing the academic rigor and readability of our manuscript. We respectfully request your review and look forward to receiving your guidance and recognition.

Comment 2: Thank you for stating the following financial disclosure:

Funding: This work was supported by the Natural Science Foundation of China under Grant [number 41561024].

Response 2: To ensure the accuracy of the information, we have revised the 'Role of Funder' statement mentioned above and hereby submit it to you through this reply letter. Meanwhile, we authorize your journal to update the corresponding content in the online submission form on our behalf. The funder (Bo Zhang) not only provided financial support but also played a significant role in conceptualization and handled the writing, review, and editing for this study.

Funding: This work was supported by the Natural Science Foundation of China under Grant [number 41561024].

Comment 3: Thank you for stating the following in the Acknowledgments Section of your manuscript:

“This research was funded by the National Natural Science Foundation of China (NO. 41561024)’”.

“This research was funded by the National Natural Science Foundation of China (NO. 41561024).”

Response 3: We would like to express our profound gratitude for your meticulous review of this manuscript and precise guidance on the placement of funding information. The issue you have identified is of paramount importance, and we fully comprehend and concur with PLOS ONE's standard requirements for the expression of funding information. This standard dictates that all funding information must exclusively appear in the "Funding Statement" section of the online submission form, and shou

---

## [Decision Letter · Decision Letter 2]

18 Jul 2025

Primary Determinants of Water Yield Services in Arid NW China: An Empirical Analysis of Gansu Province

PONE-D-25-00782R2

Dear Dr. zhang,

We’re pleased to inform you that your manuscript has been judged scientifically suitable for publication and will be formally accepted for publication once it meets all outstanding technical requirements.

Kind regards,

Pradeep Kumar Badapalli

Academic Editor

PLOS ONE

Additional Editor Comments (optional):

The revised manuscript demonstrates significant improvement and is now scientifically sound. You have addressed the reviewers’ comments thoroughly in both Revision 1 and Revision 2, enhancing the clarity, methodology, and overall presentation of the work. I appreciate your efforts and recommend the manuscript for acceptance.

Reviewers' comments:

Reviewer's Responses to Questions

**Comments to the Author**

1. If the authors have adequately addressed your comments raised in a previous round of review and you feel that this manuscript is now acceptable for publication, you may indicate that here to bypass the “Comments to the Author” section, enter your conflict of interest statement in the “Confidential to Editor” section, and submit your "Accept" recommendation.

Reviewer #1: All comments have been addressed

Reviewer #3: All comments have been addressed

2. Is the manuscript technically sound, and do the data support the conclusions?

Reviewer #1: Yes

Reviewer #3: Yes

3. Has the statistical analysis been performed appropriately and rigorously? 

Reviewer #1: Yes

Reviewer #3: Yes

4. Have the authors made all data underlying the findings in their manuscript fully available?

Reviewer #1: Yes

Reviewer #3: Yes

5. Is the manuscript presented in an intelligible fashion and written in standard English?

Reviewer #1: Yes

Reviewer #3: Yes

6. Review Comments to the Author

Reviewer #1: I reviewed the response to reviewers and the manuscript with tracked changes. It appears that all of my concerns were addressed - thanks..

Reviewer #3: Dear Authors, the manuscript is scientifically sound and meets the journal's standards. In both Revision 1 and Revision 2, the authors have clearly addressed the comments and suggestions.

7. PLOS authors have the option to publish the peer review history of their article (what does this mean? ). If published, this will include your full peer review and any attached files.

**Do you want your identity to be public for this peer review?** For information about this choice, including consent withdrawal, please see our Privacy Policy .

Reviewer #1: **Yes: ** John C. Ayers

Reviewer #3: No

---

## [Editor Report · Acceptance letter]

PONE-D-25-00782R2

PLOS ONE

Dear Dr. Zhang,

I'm pleased to inform you that your manuscript has been deemed suitable for publication in PLOS ONE. Congratulations! Your manuscript is now being handed over to our production team.

Kind regards,

on behalf of

Dr. Pradeep Kumar Badapalli

Academic Editor

PLOS ONE